# Towards the Magic Radioactive Bullet: Improving Targeted Radionuclide Therapy by Reducing the Renal Retention of Radioligands

**DOI:** 10.3390/ph17020256

**Published:** 2024-02-16

**Authors:** Kim E. de Roode, Lieke Joosten, Martin Behe

**Affiliations:** 1Department of Medical Imaging, Nuclear Medicine, Radboud University Medical Center, Geert Grooteplein Zuid 10, 6525 GA Nijmegen, The Netherlands; kim.deroode@radboudumc.nl; 2Tagworks Pharmaceuticals, Toernooiveld 1, 6525 ED Nijmegen, The Netherlands; 3Center for Radiopharmaceutical Sciences ETH-PSI-USZ, Paul Scherrer Institut, 5232 Villigen, Switzerland

**Keywords:** kidney, low-to-moderate-molecular-weight radioligands, TRT, renal retention, tumor targeting

## Abstract

Targeted radionuclide therapy (TRT) is an emerging field and has the potential to become a major pillar in effective cancer treatment. Several pharmaceuticals are already in routine use for treating cancer, and there is still a high potential for new compounds for this application. But, a major issue for many radiolabeled low-to-moderate-molecular-weight molecules is their clearance via the kidneys and their subsequent reuptake. High renal accumulation of radioactive compounds may lead to nephrotoxicity, and therefore, the kidneys are often the dose-limiting organs in TRT with these radioligands. Over the years, different strategies have been developed aiming for reduced kidney retention and enhanced therapeutic efficacy of radioligands. In this review, we will give an overview of the efforts and achievements of the used strategies, with focus on the therapeutic potential of low-to-moderate-molecular-weight molecules. Among the strategies discussed here is coadministration of compounds that compete for binding to the endocytic receptors in the proximal tubuli. In addition, the influence of altering the molecular design of radiolabeled ligands on pharmacokinetics is discussed, which includes changes in their physicochemical properties and implementation of cleavable linkers or albumin-binding moieties. Furthermore, we discuss the influence of chelator and radionuclide choice on reabsorption of radioligands by the kidneys.

## 1. Introduction

Since the introduction of the “magic bullet” concept by Paul Ehrlich in the early 1900s, researchers have been aiming to develop drugs that, upon administration, selectively reach a lesion. This concept is now one of the foundations of targeted therapies in the field of cancer treatment. Targeted radionuclide therapy, in which a radionuclide is coupled to a tumor-targeting molecule, is one of the available treatments for cancer nowadays. In the last few decades, the field of targeted radionuclide therapy has gained popularity, as illustrated by the increasing number of publications on this topic every year. Despite the successful use of several radioligands for targeted radiotherapy (TRT) in clinical application, TRT cannot be used to its full potential due to an unfavorable body distribution of many radioligands. 

Targeted radionuclide therapy aims to damage cells in cancerous tissues severely enough to undergo cell death. This mainly occurs via radiation-induced single- and double-strand breaks in the DNA, either directly or indirectly via the formation of reactive oxygen species that induce oxidative stress. As radiation damages cells within the irradiated area, targeted radionuclide therapy is an effective way to eliminate complete cancerous tissues due to the crossfire effect. On the other hand, the presence of radioactive molecules in healthy non-target organs may induce damage in these organs as well. To reduce radiation-induced toxicity in healthy organs, the radioactive payload is coupled to a tumor-targeting moiety, forming a radioligand. The tumor-targeting moieties are generally designed to bind to cell surface receptors or biomolecules that are (over)expressed in the tumor microenvironment and have no or sufficiently low expression in non-tumor tissue. Compared to untargeted radiotherapies, the tumor-seeking ability of targeted radionuclide therapies makes these therapies effective in the elimination of primary tumors, including any (undetected) metastasis. In addition, targeting may reduce side effects of the treatment as a result of the low exposure of healthy tissues. 

Since many low-to-moderate-molecular-weight radioligands are excreted and actively reabsorbed by the kidneys, this organ represents one of the dose-limiting organs for many targeted radionuclide therapies [1,2,3]. The current threshold for the absorbed kidney dose after radionuclide therapy is 23 Gy and is extended from external beam radiation [4,5]. However, in practice, different types of radiation treatment require different thresholds, as kidney tolerance depends, among others, on emission type and range, radiation energy, and dose distribution of the radiation [4]. Clinical symptoms generally develop after a latent period of several months or even years after the treatment and include hypertension, albumin urea, anemia, and other symptoms identical to renal failure of other causes. As the damage is largely irreversible, treatment of nephropathy relies on the management of the symptoms (blood pressure control, correction of metabolic acidosis), or in more severe disease renal replacement therapies such as dialysis [6].

Several potential solutions have been explored to reduce the dose delivered to the kidneys during targeted radionuclide therapy. Although multiple strategies are applied to protect the kidneys, such as dose fractionation, in this review, we will specifically focus on strategies that aim to reduce radioligand uptake in order to prevent nephrotoxicity. The strategies discussed here are: (i) choice of non-residualizing radionuclides, (ii) competitive uptake inhibition with non-radioactive substances, (iii) adjustments in the chemical design of the radioligand related to charge, sterics, and chelator choice, (iv) introduction of a cleavable linker, and (v) binding to albumin (Figure 1). Next to radionuclide therapy, the detection of tumor lesions by radionuclide imaging may also be hindered by high kidney uptake of the radioligand. However, here we will focus on radioligands used in of therapeutic studies, which we define as being (or being capable of being) labeled with therapeutic radionuclides. Among therapeutic radionuclides, we consider beta and, as more recently being explored in nuclear medicine, alpha emitters. As high kidney uptake is a problem mainly in TRT and therapies utilizing radiolabeled peptides and small proteins like Fab’ fragments, nanobodies, or DarPins, which exploit radioligands with a size below 60 kDa (glomular filtration barrier), this will be the main topic of the current review.

## 2. Kidney Retention of Low-to-Moderate-Molecular-Weight Molecules

The exact biological mechanisms underlying the accumulation of radioactivity in the kidneys are not fully identified, as evidence supports the involvement of multiple mechanisms for different radioligands [7]. However, it has been shown that the reuptake processes of radioactivity mainly take place in the renal tubules, leading to accumulation in the kidney cortex (Figure 2). As amino acids are highly valuable to the body, cells in the renal tubules have developed mechanisms enabling the reabsorption of (small) proteins and peptides so that the body can reuse them. Peptide- and protein-based radioligands are often based on endogenous biomolecules, thereby making them susceptible to uptake via these mechanisms. 

Elimination of high molecular weight and lipophilic compounds generally occurs via the hepatobiliary route [8]. Compounds are taken up by hepatocytes (either actively or passively), where, depending on their structure, they can undergo metabolism to a more hydrophilic metabolite. Metabolic end products are secreted in bile, enabling elimination via the intestine, or in the blood, allowing elimination via urine. Most peptide-based radioligands are small and hydrophilic and are therefore directly cleared via the urinary route. Hydrophilic, non-protein bound radioligands circulating in the blood reach the kidneys via the renal artery and subsequently are filtered through the glomerulus as long as their size is small enough to pass through the glomerular filtration barrier (about 60 kDa) [9] (Figure 3). Via this route, the radioligand reaches the ultrafiltrate traveling through the tubules. The proximal tubule is lined with proximal tubule cells rich in transporters, which enable reabsorption of biomolecules and water from the pre-urine. Although multiple transporters are involved in this process, the megalin/cubilin complex plays a major role. These proteins are abundantly expressed on the apical membrane of the proximal tubule and facilitate reabsorption of a diversity of biomolecules, including protein-based radioligands [9,10]. Megalin is a transmembrane receptor member from the low-density lipoprotein receptor superfamily. The protein has a large extracellular part consisting of multiple different cysteine-rich binding domains responsible for binding structurally distinct biomolecules [11]. Cubilin is also involved in binding biomolecules, but as this protein lacks the transmembrane domain needed for internalization, it relies on complexation with, among others, megalin for internalization and recycling [12]. Upon binding of a radiolabeled molecule, the megalin/cubilin receptor–radioligand complex internalizes to an endosome in which the molecule dissociates in a pH-dependent manner. Directed to a lysosome, where enzymes are responsible for the breakdown of the radiopeptide, eventually yielding the biomolecule’s monomeric units, such as amino acids, these biological fragments are transported back to the blood via transporters on the basolateral side of the proximal tubule cells. The radiometal–chelate is retained inside the tubule cells, exposing the tubule cells and their neighboring cells to extended radioactivity [2].

In addition to this rather non-specific uptake mechanism in the renal tubules, some radioligands are taken up by their target protein, which is expressed in the kidneys. For example, the targeting of neuroendocrine tumors by somatostatin analogues is challenged by somatostatin receptor subtypes expressed in the glomeruli, proximal and distal tubules, and loop of Henle in human and murine kidney [13,14]. Uptake via these receptors also contributes to the uptake of radiolabeled somatostatin analogues, such as radiolabeled octreotide derivatives. About 20% of the kidney uptake with OctreoScan is related to the somatostatin receptor expression in the kidney [15]. 

This is also the case for ligands binding PSMA (prostate-specific membrane antigen), which is overexpressed in prostate cancers. PSMA is a highly promising target and has been explored since the beginning of the century with targeting antibodies and later with small compounds based on the Lys-urea-Glu motif as radioconjugates. The efforts resulted in the approval by the Food and Drug Administration (FDA) and European Medicines Agency (EMA) of a positron emission tomography (PET) tracer and therapeutic radiolabeled compound for the diagnosis ([^68^Ga]Ga-PSMA-11) and therapy ([^177^Lu]Lu-PSMA-617) of metastatic castration-resistant prostate cancer (mCRPC). In addition to expression in tumor lesions, the expression of PSMA in the brush borders and apical cytoplasm of a subset of proximal tubules has been reported [16], and in mice, a nearly 100% contribution to the uptake of radioactive PSMA ligands has been shown [17]. Also, glucagon-like peptide-1 receptors (GLP-1R) have been detected in the kidneys of rodents [18], pigs [19], and humans [20]. However, several reports indicate that in humans, the receptor is solely expressed in smooth muscle cells rather than in proximal tubules [21,22], thereby not contributing to the specific reabsorption of radiolabeled exendin analogues in the kidneys. In addition, while cholecystokinin receptors (CCKR) have been associated with multiple tumor types, the expression of CCKR type 1 receptors (CCK1R) was described in the kidneys of rodents [23] and humans [22], and cholecystokinin type 2 receptors (CCK2R) were detected in the tubules of rodents [21,23,24]. It remains elusive, however, if human tubule cells also express GLP-1R and CCKR, which may contribute to the uptake of radiolabeled exendin and gastrin or cholecystokinin analogues.

## 3. Radionuclide-Dependent Kidney Uptake Reduction

One approach to reduce the accumulation of radioactivity in the tubular kidney cells is the use of different radionuclides. Although the usefulness of this approach is often reflected in the relatively low kidney uptake of diagnostic nuclides like Tc-99m and F-18, we focus in this section on therapeutic nuclides. The group of Lahoutte et al. used this approach to reduce kidney uptake of Camelid single-domain antibody-fragment (sdAb) 2Rs15d targeting Her2/neu, the human epidermal growth factor receptor 2, which plays a major role in breast cancer and other entities. When radiolabeling these internalizing nanobodies, which are small in size (15 kDa), with radiometals such as Lu-177, renal uptake is too high for therapeutic use, even in combination with different kidney uptake reduction approaches. Therefore, they decided to use the therapeutic non-radiometallic radionuclide I-131. The sdAb targeting Her2/neu was radiolabeled with I-131 via the tumor residualizing prosthetic group *N*-succinimidyl 4-guanodinomethyl-3-[^131^I]I-benzoate ([^131^I]I-SGMIB), which was introduced into the field by Zalutsky in 2007 [25]. The prosthetic group [^131^I]I-SGMIB was specifically designed to have low uptake in the kidneys but high retention in the tumor cells. The low kidney uptake is a result of the formation radiocatabolites that have minimal tubular reabsorption leading to rapid excretion, presumed to be an effect of charge [25]. In contrast, good retention in the tumor cells after receptor-mediated internalization of the protein occurs as a result of the high pKa of the guanidino group that causes entrapment of the charged radiocatabolites after lysosomal degradation [25,26].

^177^Lu- and ^131^I-radiolabeled 2Rs15d were examined in Her2/neu positive tumor models in mice. The group used the subcutaneous ovarian cancer cell model with SKOV3 cells to examine the ^177^Lu-labeled compounds and the subcutaneous breast cancer BT474/M1 model for the ex vivo biodistribution of the ^131^I-labeled compounds. The results are therefore comparable to a limited extent. The iodinated sdAb showed a higher tumor uptake after 3 and 24 h of 17.7 and 5.1 %IA/g, compared to the [^177^Lu]Lu-2Rs15d combined with Gelufosine—a kidney uptake reduction substance that is described in detail in Section 4.2, which showed 5.97 and 3.58 %IA/g, respectively. The clearance from the kidney was much faster for [^131^I]I-2Rs15d, which was reduced from 12.5 %IA/g after 3 h to 0.94 %IA/g after 24 h, whereas it was 7.59 and 4.33 %IA/g for [^177^Lu]Lu-2Rs15d. This resulted in a favorable dosimetry for [^131^I]I-2Rs15d. The highest dose injected (37 MBq) was delivered to the tumor with 11.9 Gy, whereas the kidneys received 8.4 Gy. The doses to other organs were very low. A therapeutic application, where BT474/M1 xenografted mice receiving ~10 MBq [^131^I]I-2Rs15d, showed a significantly longer median survival of 137.5 days versus 93.5 and 78 days for mice receiving [^131^I]I-control sdAb and vehicle, respectively [26]. Based on these results, [^131^I]I-2Rs15d was transferred to a clinical study for dose estimation in six healthy volunteers, followed by a first evaluation in three patients with a Her2/neu positive tumor. An acceptable dose delivery of 1.54 ± 0.25 mGy/MBq to the kidneys was observed, and to bone marrow, it was 0.03 ± 0.01 mGy/MBq. SPECT/CT (single photon emission computed tomography/computed tomography) images in patients with advanced breast cancer showed focal uptake of [^131^I]I-2Rs15d in metastatic lesions [27]. A multi-center dose escalation study is ongoing to evaluate safety, tolerability, dosimetry, and preliminary efficacy of the HER2/neu directed radioligand CAM-H2 ([^131^I]I-2Rs15d) in patients with advanced/metastatic HER2/neu-positive breast, gastric, and gastroesophageal junction cancer (NCT04467515).

Utilizing non-residualizing radioisotopes with improved tumor retention is one of the rare successful methods to achieve a therapeutic applicable small protein for targeted radionuclide therapy with a tolerable dose delivered to the kidneys. However, efficacy has yet to be demonstrated in clinical trials.

## 4. Competitive Inhibition in Proximal Tubules

As the main reason for the high uptake of radioligands in the kidneys is their tubular reabsorption via proteins expressed on the proximal tubule cells, the co-administration of non-radioactive ligands can inhibit renal uptake by competing for these proteins. As already discussed, megalin is such a well-known receptor involved in the absorption process of many structurally different proteins and peptides. As this suggests it to be a nonselective transport system, it is not surprising that a wide variety of ligands are able to inhibit the renal uptake of several ligands. The involvement of megalin in the reabsorption process has been demonstrated for many (radiolabeled) ligands. In megalin-deficient mice (mice with kidney-specific inactivation of the megalin gene), the renal uptake was significantly lower compared to wild type mice. This has been reported for several hormones and vitamins, as well as radiolabeled octreotide, minigastrin, and anti-EGFR nanobody [10,28,29,30].

An important aspect of this competitive strategy is ensuring that these ligands do not interfere with the binding of the radiolabeled compound to their receptor on target lesions, as this would result in an undesired, decreased uptake as well. A few of these competitors will be discussed in the next section, attempting to provide an impression on the progress that has been made in translating from diagnostic to therapeutic agents. 

### 4.1. Charged Amino Acids

Perhaps the most well-known renoprotective strategy is the co-infusion of positively charged amino acids, like lysine and arginine, which compete for the megalin and/or cubilin receptors on the proximal tubule cells. Early studies showed a majorly reduced kidney uptake of an ^111^In-labeled Fab fragment in mice [31] or ^111^In-labeled somatostatin analogues in rats [32] or human [33] after infusion of lysine. Many successful stories have followed, mainly focusing on the reduced renal uptake of somatostatin analogues after infusion of lysine (and arginine). These studies include [^111^In]In-DTPA0-octreotide in human (33 ± 23% inhibition) [34], but also, therapeutic compounds, such as [^177^Lu]Lu-DOTA^0^,Tyr^3^]-octreotate [35] and [^161^Tb]Tb-DTPA-octreotide [32], demonstrated lower renal retention in rats. Furthermore, as the field of alpha-emitting radionuclides is evolving at a breakneck pace, it is of utmost importance that the inhibition of renal uptake is as efficient for these radioligands. [^212^Pb]Pb-DOTAMTATE [36] and [^213^Bi]Bi-DOTATATE [37] in mice showed that kidney uptake greatly reduced by the co-administration of amino acids. Furthermore, kidney-absorbed dose was reduced by half after receiving lysine [37]. In addition, patients with differentiated neuroendocrine tumors who showed progressive disease after treatment with beta-emitters were offered a new treatment with the alpha-emitter [^213^Bi]Bi-DOTATOC. Moderate chronic toxicity was reported, even after a mixture of lysine, arginine, and Gelofusine was co-infused, albeit after the kidney tolerance dose had already been reached during the earlier treatment with beta emitters, co-infused with amino acids for kidney protection [38]. Nowadays, co-infusion of amino acids (L-lysine and L-arginine) is a standardized regimen for kidney protection in TRT with [^177^Lu]Lu-DOTA-TATE (Lutathera) [39]. Interestingly, Gotthardt et al. showed in rat studies that inhibition by lysine was not effective for other radiolabeled peptides besides octreotide. On the contrary, polyglutamic acid (PGA) was able to inhibit renal uptake of minigastrin and exendin, whereas it did not lower kidney uptake of octreotide [7]. The concept of inhibition by PGA, which consists of negatively charged amino acids, was first demonstrated in mice by Béhé et al. [40]. Megalin possesses four large binding domains [11], which might be the reason why renal uptake of octreotide can be blocked by lysine and renal uptake of minigastrin can be blocked by PGA. It is plausible that if the peptide binds to a distinct binding site, that binding site could be blocked by a different substance [7,10]. Also, the affinity of the peptide for the binding site may play a role in how efficient renal reabsorption can be inhibited. 

### 4.2. Gelofusine

In the first reports on succinylated gelatin plasma expanders (e.g., Gelofusine), an increased urinary excretion of low-molecular weight compounds, also known as proteinuria, has been described [41]. Subsequently, the effect of Gelofusine on the renal accumulation of radiolabeled peptides was studied. Gelofusine consists of several components, which differ in structures and sizes and are mostly neutral or negatively charged proteins, but the mechanism by which this plasma expander inhibits kidney uptake is not fully understood, although the endocytic receptor megalin most probably plays a role [42,43]. Gelofusine was shown to efficiently reduce renal uptake of ^111^In-octreotide in healthy mice and rats [43], without affecting uptake in somatostatin-expressing tumors [44]. Gotthardt et al. demonstrated that Gelofusine not only inhibited the renal reabsorption of ^111^In-labeled octreotide but of exendin, minigastrin, and bombesin as well [7]. Other radioligands for which Gelofusine has proven to efficiently reduce kidney uptake in mice (with reductions of 50–70%) include [^111^In]In-CP04 [45], [^111^In]In-DOTA-RAFT-RGD (targeting α_v_β_3_ integrin) [46], and [^64^Cu]Cu-cyclam-RAFT-c(-RGDfK-)4 [47], and even a fluorescently labeled compound, namely A700-RAFT-RGD [46]. In all these studies, the administration of Gelofusine did not negatively influence tumor uptake but led to better tumor-to-kidney ratios. Next to reducing kidney uptake of radiolabeled peptides, Gelofusine also has been shown to be beneficial for the pharmacokinetics of nanobodies. Kidney uptake of the anti-EGFR [^99m^Tc]Tc-7C12 nanobody was inhibited by 36% when Gelofusine was co-administered [30]. Furthermore, the feasibility of Gelofusine has been confirmed in clinical studies. A reduced renal uptake of [^111^In]In-octreotide by 45% has been presented, which is necessary for the safe use this compound for TRT [48]. For [^111^In]In-exendin-4 (reduction of 18%) and [^68^Ga]Ga-NOTA-exendin-4 (reduction of 57%), the advantage of reduced renal uptake is twofold; it may not only improve the visualization of insulinomas close to the kidneys [49,50], but it is also indispensable for enabling TRT using radiolabeled exendin. Gelofusine is able to inhibit the renal uptake of several structurally different molecules, which may be explained by the mixture of components that Gelofusine consists of. These components may bind to more than one binding site and therefore are able to inhibit the reabsorption of a variety of compounds.

Whereas these studies were primarily designed to demonstrate the feasibility of Gelofusine, few studies using Gelofusine in combination with a therapeutic radioligand have been documented. The kidney uptake of the peptide ^213^Bi-P-P4D, which targets the urokinase-type plasminogen receptor (uPAR), was reduced by half in a mouse model bearing human ovarian tumors [51]. Also, for the 2Rs15d-nanobody, labeled with either Lu-177, Ac-225, or Bi-213, a reduction in kidney uptake of more than 50% has been presented [52,53,54].

Despite the promising biodistribution data, only a few studies investigated if the effect of Gelofusine is enough to deliver a treatment dose to the tumor without causing damage to the kidneys. Buitinga et al. addressed this using dosimetry in a clinical study with In-111-labeled exendin and estimated that an effective treatment tumor dose with Lu-177-labeled exendin could be administered safely [49]. Dhuyvetter and co-workers showed that treating mice with [^177^Lu]Lu-DTPA-2Rs15d did not reveal any visible toxicity to the kidneys, but how this translates to human was not mentioned [52]. Conversely, the use of Gelofusine in combination with [^225^Ac]Ac-DOTA-2Rs15d or [^213^Bi]Bi-DTPA-2Rs15d was not enough to prevent renal toxicity, and therefore, further optimization is needed before this compound can be applied safely [53,54,55].

Several studies showed that a combination of inhibitors, such as Gelofusine and lysine, or PGA and Gelofusine, had an additive effect on the inhibition of renal reabsorption [7,30,44,56]. This implies that one radioligand can be absorbed via different uptake mechanisms or that the radioligands bind to different binding domains of megalin. Since side effects have been reported for both the use of amino acids (e.g., vomiting, nausea, and hyperkalemia) [48], as well as for Gelofusine (e.g., allergic reactions) [35], it needs to be examined to what extent the side effects are when a combination of these inhibitors is applied.

## 5. Chemical Design of Radioligand

Uptake of radioactive compounds in the renal system depends strongly on the physicochemical properties of the radioligand, among others resulting from the negative charge of the renal brush border and the charge of the proximal tubule cell surface. Also, a ligand may undergo metabolism and/or chemical instability once administered or internalized into tumor and/or tubular cells, influencing the accumulation of radioactivity in the kidneys. Resultingly, numerous attempts have been made to adjust the design of a radioligand in order to optimize pharmacokinetics and limit the retention in the kidneys. This section will touch upon several strategies applied to radioligands and discuss the influence of specific modifications on renal uptake, as established in small animal models. As targeting peptides are often thoughtfully designed in order to obtain high-affinity binding to a target molecule, we focus on changes implemented in either the chelating moiety (stability, residualizing activity, and charge) or the linker (charge, sterics, length, and type). Figure 4 summarizes the strategies related to the chemical design of a radioligand discussed in this section.

### 5.1. Influence of Chelator

Depending on the radiometal planned to be used for the intended purpose of the radioligand, several different chelators may be chosen for efficient coordination of the metal. Figure 5 displays the different chelators discussed in this chapter. Aiming for a radioligand with ideal pharmacokinetic properties, ample attempts have been made to improve pharmacokinetics by chelator substitution. Due to the lack of biodistribution studies on chelator variations in radioligands labeled with therapeutic nuclides, examples of diagnostic radionuclides are also discussed, including chelators with the potential of being labeled with therapeutic radionuclides.

#### 5.1.1. Stability

Stability of a radiometal–chelator complex is a fundamental determinant affecting the accumulation of radioactivity in the kidneys. Dissociation of radiometals from the chelator of a radioligand can contribute to renal accumulation of radioactivity. For example, the Cu-DOTA complex is susceptible to in vivo dissociation of Cu^2+^ and/or its reduced form Cu^+^ from the chelator. Subsequent fixation of the free Cu to copper-binding proteins may result in an altered biodistribution profile of the radiometal [57,58]. Resultingly, clearance route and/or kinetics may also change, influencing the radiation reaching the kidneys. It was demonstrated that substitution of DOTA for a more stable NOTA chelate with Cu-64 in a cyclic αMSH analogue led to a decrease in kidney uptake, while drastically improving tumor uptake [59]. Also, for a bombesin analogue, replacement of a [^64^Cu]Cu-DOTA complex with a more stable [^64^Cu]Cu-CB-TE2A (1,4,8,11-tetraazabicyclo[6.6.2]hexadecane-4,11-diacetic acid) led to reduced retention of radioactivity in the kidneys and an enhanced tumor-to-kidney ratio [60].

#### 5.1.2. Residualizing Activity

Some radionuclides, including Tc-99m and Re-188, have the advantage of being capable of being chelated by peptide-based chelators. When taken up and metabolized in the kidneys, these chelators produce non-residualizing catabolites, which avoid entrapment in the proximal tubule cells [61,62]. However, most therapeutic radionuclides, including Lu-177, Cu-67, and Y-90, require residualizing chelators for complexation. As a consequence, other strategies to reduce renal accumulation are needed for radioligands labeled with these isotopes.

#### 5.1.3. Charge

The interaction between a chelator and its radiometal is generally based on electrostatic interaction between the positively charged metal and polar groups of the chelator. Radiometal–chelator choice can thus influence the overall charge of a radioligand molecule and thereby affect its pharmacokinetic properties, including uptake in the kidneys. For example, in a ^68^Ga-labeled CXCR4 receptor-targeting LY2510924 antagonist, substitution of the DOTA chelator with a DOTAGA (1,4,7,10-tetraazacyclododececane-1-(glutaric acid)-4,7,10-triacetic acid) chelator (with one additional carboxylic acid) led to a significant increase in kidney uptake while decreasing tumor uptake at 1 h post-injection (p.i.) [63]. A head-to-head comparison between [^68^Ga]Ga-octreotide conjugates coupled to DOTA (three carboxylic acids), NOTA (1,4,7-triazacyclononane-1,4,7-triacetic acid) (two carboxylic acids), or NOTA with one to three additional carboxylic acid groups, also showed distinct kidney uptake and washout profiles depending on the chelator. Where [^68^Ga]Ga-DOTA and [^68^Ga]Ga-NOTA-conjugated radioligands had the lowest kidney uptake, a higher uptake was observed for radioligands coupled with [^68^Ga]Ga-NOTA with one and two additional carboxylates [64]. A study on bombesin analogues comparing conjugates with NOTA, NODAGA (1,4,7-triazacyclononane-1-(glutaric acid)-4,7-diacetic acid), DOTA, and DOTAGA labeled with In-111 revealed the highest uptake in kidneys, as well as in the tumor for the NODAGA variant at 4 h p.i. Here, the DOTA variant yielded the highest tumor-to-kidney ratios. In absolute values, the NODAGA variant had the highest kidney uptake, followed by NOTA, DOTAGA, and DOTA [65]. The same molecule labeled with Ga-68 yielded the lowest kidney uptake (and highest tumor-to-kidney ratio) for the NOTA conjugated variant at 2 h p.i., as hypothesized to be a result of the local positive charge of the metal–chelator complex [66]. Mentioned examples underline the significant influence that chelators of different charges can have on kidney accumulation of a radioligand.

However, not always is the effect of chelator charge impact kidney uptake to the same extent. In the earlier-mentioned [^68^Ga]Ga-octreotide, more than three carboxylates on the chelator were not beneficial for lowering kidney uptake, and this led to reduced tumor uptake instead [64]. In addition, for a PSMA-radioligand, little difference between [^68^Ga]Ga-DOTA and [^68^Ga]Ga-NOTA was observed in terms of tumor and kidney uptake, while [^68^Ga]Ga-HBEDCC led to significantly higher tumor uptake, accompanied by increased accumulation and slower clearance from the kidneys [67]. For a gastrin analogue, no significant difference in kidney uptake was observed between a [^68^Ga]Ga-NOTA and [^68^Ga]Ga-NODAGA variant, while tumor uptake was higher for the NOTA variant [68].

Surprisingly, comparing two conjugates of [^68^Ga]Ga-octreotide analogues with three carboxylic acid groups on the chelator (DOTA and NOTA-carboxylate) showed a kidney uptake that was twice as high for the NOTA-carboxylate conjugate than the DOTA conjugate at 30 min and 1 h p.i. [64]. A comparable observation was made for an ^111^In-labeled bombesin analogue, in which a NODAGA variant had a kidney uptake almost three times as high as the DOTA variant [65]. This implies that more factors than overall charge alone are responsible for the different kidney uptake.

A head-to-head comparison of an ^111^In-labeled gastrin analogue containing either a DOTA or DOTAGA chelator showed no significant difference in kidney uptake and tumor targeting, despite the additional negative charge. This is in contrast to when this negative charge was incorporated in the linker between the binding motive and chelator, as discussed in Section 5.2 [69]. This implies that rather than the overall charge of a radioligand alone, local charges of a radioligand may also have an influence on accumulation in the kidneys, complicating accurate prediction of in vivo behavior [66].

### 5.2. Influence of Linker Charge

Adapting the linker connecting the targeting motif to the radiometal chelate has been widely explored to improve target binding affinity and pharmacokinetics. In general, it is acknowledged that charges can have a positive influence on the renal clearance of radioligands [7,70,71,72,73]. However, direct translation between molecular charge and renal uptake remains challenging, as the exact factors determining uptake by the kidney remain unclear and are specific per radioligand. Therefore, here, we discuss several structural radioligand modifications related to charge that have influenced renal uptake in order to delineate insights gained regarding the influence of molecular charge on renal uptake of potentially therapeutic radioligands.

Aiming to reduce electrostatic interaction between positively charged peptides and the negatively charged surface of renal tubule cells, negatively charged amino acid residues can be introduced into a radioligand. Among the first to show this was the group of Akizawa et al., who in 2001 reported the influence of substituting the N-terminal amino acid of ^111^In-labeled DTPA-octreotide derivatives. The study included substitution of N-terminal phenylalanine (neutral) for aspartic acid (anionic). Evaluation in (tumor-free) mice revealed that radioactivity in the kidneys was roughly 2-fold lower for the negatively charged linker compared to the neutral one [74]. After this success, more successful implementations of negative charges in linkers of octreotide analogues followed, which significantly reduced renal accumulation compared to neutral or positive charges. These linkers include substitutions to aspartic acid [74,75], glutamic acid, γ-carboxy-glutamic acid [76], or p-carboxy-phenylalanine residues. However, not only was uptake in the kidneys reduced as a result of the substitution, but the binding affinity for the somatostatin receptor also deteriorated once the N-terminal amino acid residue of the binding motive was not aromatic [77]. In addition, it was found that the major radiometabolite, [^111^In]In-DTPA linked to the N-terminal amino acid, was less retained in kidneys once this N-terminal amino acid was lipophilic. An octreotide analogue with high tumor and low kidney retention was eventually found in the form of a variant with a D-Phe-Asp-D-Phe linker between [^111^In]In-DTPA and octreotide binding motif [75,78].

Similarly, for targeting the melanocortin-1 receptor (MC1R), a target associated with skin cancer, using cyclic melanocyte stimulating hormone (α-MSH) analogues, it was found that the insertion of a glutamic acid residue linking a DOTA (1,4,7,10-tetraazacyclododecane-1,4,7,10-tetraacetic acid) chelator and targeting peptide was effective in reducing radioactivity uptake in the kidneys in mice, while maintaining or even improving uptake in the tumor [79,80,81]. Interestingly, incorporation of two aspartic acids on the C-terminal region of a linear α-MSH variant, rather than in the linker region, did not lead to reduced uptake in the kidneys [82]. Improved tumor-to-kidney ratios were also observed after the insertion of glutamic acid into cyclic RGD peptide analogues [83], although for a different RGD radioligand analogue, the effect on kidney uptake was not significant [84]. In addition to substituting positive or neutral linkers with negatively charged linkers, substitution of glutamic acid into cysteic acid may also be an effective strategy for reducing kidney uptake, as shown for CXCR4 receptor-targeting LY2510924 antagonists. While the influence on affinity and tumor uptake was minimal, kidney uptake was significantly lower for the trisulfonic acid radioligand variant compared to the triglutamic acid variant. Potentially, this is caused by the lower pKa of the sulfonate, which increases the capacity to retain its negative charge in vivo [63].

For a gastrin analogue, it was shown that reducing the number of histidine residues between DOTA and the receptor binding motif from 6 to 2 repeats led to an almost 2-fold lower kidney uptake, while even improving tumor uptake, as measured at 4 h after administration. While at a physiological pH of 7.4, the cyclic amine of histidine residues is uncharged (pKa of 6.0), the pH of urine in the proximal tubules may be more acidic, potentially resulting in positively charged histidine side chains [85]. Therefore, it remains elusive if the observed reduction in kidney uptake is an effect of charge alone.

With the aim of reducing liver uptake of affibodies (14 kDa) to direct their excretion from the liver to the kidneys, the histidine–glutamic acid (HE) tag was designed by Hofstrom et al. [71]. For several radiolabeled ligands, including affibodies and octreotate, this tag led to a higher accumulation in the kidneys [70,71], whereas for PSMA-targeted radioligands, the tag showed to be effective in the reduction of kidney uptake, as shown in the context of diagnostic radioligands. While leaving tumor uptake for the radiolabeled ligand high, introduction of the HEHEHE tag or substitution of a neutral linker into the tag resulted in a factor 2- to 3-fold improvement of the tumor-to-kidney ratio at 1 h p.i. in mice [70,86]. This improvement was not observed when substituting the peptide linker for a more lipophilic tryptophan–glutamic acid dipeptide repeat [86]. The histidine–glutamic acid dipeptide repeat was also effective in the reduction of kidney uptake for a PSMA ligand–bombesin analogue peptide heterodimer connected to an HBED-CC (N,N′-bis-[2-hydroxy-5-(carboxyethyl)benzyl]ethylenediamine-N,N′-diacetic acid) chelator labeled with Ga-68, as demonstrated by a reduction in the area under the curve (AUC) of more than 50% in the kidneys compared to the radioligand without the charged linker [87].

The effect of adding positive charges on kidney uptake was determined for cyclic α-MSH analogues by insertion of a cationic piperidine linker between the chelator and targeting peptide. As a result of the high pKa of the piperidine unit, the linker has a cationic charge at physiological and urinary pH. However, minimal differences were observed in kidney uptake between variants with a neutral polyethylene glycol 2 (PEG2) linker, a single piperidine linker, and a variant with two piperidine units, demonstrating the small influence that a cationic charge can have on renal uptake of these ligands [88]. In contrast, an earlier mentioned study on octreotide analogues demonstrated a large influence after the substitution of N-terminal phenylalanine for cationic lysine. In this study, the renal uptake of mice was more than 2-fold higher for the cationic linker variant, as measured up to 6 h after administration [74].

### 5.3. Influence of Amino Acid Sterics

In addition to electrostatic interaction, receptor–ligand binding depends on the fit of the three-dimensional structure of the ligand into the ligand-binding region of the receptor. As mentioned earlier, significant absorption of peptides is receptor-dependent. It is therefore not surprising that steric changes in a radioligand may influence uptake in the tubule cells. It was demonstrated that for CCK2R targeting on tumors, the kidney uptake of gastrin analogues strongly reduced when a pentaglutamic acid between the target binding motive and [^111^In]In-DOTA chelator was deleted or shortened [69,85,89]. Rather than being a result of the different charge of the molecule, this revealed an effect of sterics, since changing the penta L-glutamic acid sequence into D-glutamic acids led to a 90% reduction in kidney uptake, while maintaining the same molecular charge. Interestingly, the change did not significantly reduce uptake in the tumor [89]. Substitution of L-glutamic acids into D-glutamic acids, however, does not always contribute to a more favorable kidney retention of a radioligand, as demonstrated by CXCR4 targeting LY2510924 antagonists. Here, substitution of an L-triglutamic acid sequence to a D-triglutamic acid did not reduce radioligand uptake in the kidneys [63].

In addition to stereochemical variations, the size of amino acid side chains may also influence uptake in the renal system. Recently, it was demonstrated in mice that for a ^68^Ga-labeled PSMA radioligand, replacement of the glutamic acid (two CH2) in the Lys-urea-Glu PSMA-binding motive by an Asp (one CH2) or Aad (three CH2) strongly reduced kidney uptake while retaining high tumor uptake. Tumor-to-kidney ratios were improved 21.6 and 34.6-fold, as measured 1 h after injection [90]. More successful adaptations in the structure of a PSMA ligand were performed by substitution of the naphtylalanine residue between the PSMA binding motive and DOTA chelator labeled with Ga-68. Substitution into 2-indanylglycine gave a slight reduction in kidney uptake, but more profound was the reduction of kidney uptake of the 3,3-diphenylalanine linker variant. Despite a small reduction in tumor uptake, this radioligand improved the tumor-to-kidney uptake more than 3-fold compared to the naphtylalanine linker variant via an unknown mechanism [91]. For CXCR4 receptor targeting LY2510924 antagonists, a slightly lower kidney uptake but lower tumor uptake was also observed for substitution of the triglutamate to one carbon shorter side chain variant tri-aspartate, while only slightly reduced tumor uptake was seen for the one carbon longer side chain tri-homoglutamate variant [63]. For a bombesin analogue, a significantly higher accumulation in the kidneys was observed for an aromatic glycine-p-aminobenzoic acid linker compared to a linear 8-aminooctanoic acid linker and the p-aminobenzoic acid moiety alone [92]. Since the discussed steric adaptations and their effects on radioligand uptake in the kidneys are very radioligand-specific, the difference in kidney uptake is feasible to be an effect of reduced interaction with specific receptors in the renal tubule that interact with specific domains of the radioligand rather than non-specific transporters.

### 5.4. Influence of Linker Length and Type

Where linkers are often introduced to a radioligand to bridge the link between the target binding motive and radiometal chelate while retaining target binding, the choice of linker may also influence the accumulation of a radioligand in the kidneys. Linkers can influence binding to transporters in the tubules via steric effects or influence the lipophilicity of the overall compound, thereby altering pharmacokinetics.

PEG is a widely used linker unit between a pharmacophore and a chelator. Due to its flexibility and high solubility, PEG can contribute to favorable characteristics of a radioligand. As PEG is able to coordinate water molecules, the introduction of a PEG linker increases the hydrodynamic size of a ligand, which may alter its route of clearance and circulation time [93]. Several research groups have investigated the influence of incorporating a PEG linker on the pharmacokinetic properties of a radioligand. For example, incorporation of a relatively high molecular weight PEG linker of 3400 Da, separating a cyclic RGD dimer from a [^64^Cu]Cu-DOTA complex, led to enhanced clearance from blood circulation and a significantly lower uptake in the kidneys, as measured up to 4 h p.i. [94]. For the corresponding RGD monomer, PEGylation yielded a higher radioactive signal from the kidneys at 30 min p.i., resulting from enhanced blood clearance, while uptake at 4 h was significantly lower than the unPEGylated variant [95]. On the other hand, a somewhat higher accumulation in the kidneys was observed upon incorporation of a high molecular weight PEG linker of 3500 Da in between a bombesin analogue and ^64^Cu-labeled DOTA compared to the same molecule with a small alkyl linker at 4 h p.i., but no significant difference was detected at 2 and 24 h p.i. [96].

Incorporation of relatively small PEG linkers on renal uptake seems to be less effective. For a bombesin analogue, investigation of the influence of a PEG linker up to six units demonstrated little influence of the PEG linker length on radioactivity accumulation in the kidneys of mice [97,98]. Also, for cyclic RGD analogues, no significant influence was observed between a variant with and without a PEG4 linker between the binding peptide and DOTA [84], and subtly lower kidney uptake at 4 h p.i. for a double PEG2 linker (amide linked) of cyclic RGD [99]. This study also examined the influence of a galactosidase linker, a strategy effective for pharmacokinetic optimization of diagnostic RGD radioligands [100]. Compared to the radioligand variant without a linker, introduction of the PEG4-galactose-based linker led to a reduced kidney uptake of almost a factor of two; however, at the cost of lower tumor uptake [99].

Also, relatively small lipophilic alkyl linkers can influence the uptake in the renal system. For bombesin analogues, it was demonstrated that an alkyl linker of a size of 11 hydrocarbons led to slightly enhanced kidney uptake compared to variants with smaller or no linker in between the chelate and peptide, as measured at 1 h p.i. [101].

In summary, subtle changes in chemical design can drastically influence the accumulation of a radioligand in the kidneys and in other (non-)target organs, as structural changes may affect receptor affinity, stability, clearance kinetics, and/or route. Effects are generally difficult to predict; an effective formula for one radioligand does not necessarily lead to the same results for another.

## 6. Cleavable Linkers

A further promising strategy for the reduction of kidney uptake of (therapeutic) radioligands is the introduction of a cleavable linker between the radionuclide and the targeting moiety. The concept is that the linker is cleaved by enzymes on the proximal tubular brush border membrane in the kidney, which separates the radioactive moiety from the targeting moiety, resulting in a radioactive metabolite that is not processed by kidney reuptake mechanisms and is directly excreted in the urine. The cleavable linkers discussed in this section are summarized in Table 1.

The idea to use a similar concept already occurred in the 90s for antibody fragments for radioimmunoimaging. Antibody fragments are used because of their better diffusion potential compared to antibodies. However, these fragments suffer from faster whole-body clearance by the kidneys, resulting in lower tumor uptake as well as dramatically higher kidney uptake. The groups of Dean et al. and Corstens et al. made the first attempts to reduce the kidney uptake of ^99m^Tc-radiolabeled antibody fragments (Fab’). They compared different chemical coupling moieties like esters or amides in combination with the thiol containing chelator mercaptoacetylglycylgycylglycin (MAG3) for Tc-99m, which is known from clinical application to have fast kidney clearance. Esters tend to hydrolyze in vivo. Dean et al. showed that the ester binding was superior to the amide binding in respect to kidney uptake reduction. They achieved a reduction of kidney uptake from about 45 %IA/g of the conventionally coupled fragment to about 8 %IA/g for the ester linkage [102]. Corstens et al. evaluated the ester coupling to two different Fab’ and could show similar results for kidney uptake reduction. They additionally explored tumor uptake, which was not significantly changed, resulting in a significantly improved tumor-to-kidney ratio, which may allow better diagnostic imaging with a lower dose burden for the patient [103].

A similar approach where hydrazinonicotinic acid (HYNIC) as a chelator for Tc-99m was covalently appended to the sulfhydryl groups of the protein via a thioether or disulfide group in combination with ester and/or amide linkers between HYNIC and Fab’ failed and resulted in higher kidney uptake compared to the conventional directly coupled chelator [104].

Arano et al. introduced the idea of using a renal enzymatically cleavable linker. They tested the glycyl-lysine sequence, which is a substrate for one of the brush border enzymes, carboxypeptidase M [105]. Therefore, they synthesized three derivatives of radioiodinated meta-iodohippuric acid, which are highly stable against deiodination, in combination with glycyl-lysine as a linker. The proof of concept was done with a Fab’ targeting osteogenic sarcomas. They showed that all three linker derivatives had a significantly better kidney-to-blood ratio in comparison to radiolabeling without the glycyl-lysine linker ([^125^I]I-Fab). The best results were achieved with the 3′-[^131^I]Iodohippuryl Nε-maleoyl-L-lysine ([^131^I]I-HML) linker. This compound was tested in tumor-bearing mice and was compared to [^125^I]I-Fab. The kidney uptake was reduced from ~9 %IA/g to ~3 %IA/g, whereas the tumor uptake was retained at ~10 %IA/g. This resulted in a significantly better tumor-to-kidney ratio of 5.0 compared to 1.3 for the control, whereas the tumor-to-blood ratios were similar (~3) [106]. Subsequently, this group established a test system to test the cleavable linker in vitro. Therefore, they isolated renal brush border membrane vesicles according to a protocol they had developed [107]. The compounds were tested in vitro on these vesicles for their cleavability. A further attempt to optimize the linker by replacing the lysine with other moieties was not successful and showed no better results [108]. An attempt to transfer this approach to radiometal chelates exemplified with Cu-64 NODAGA with exendin-4 failed and showed no lower kidney uptake [109].

The use of a Meprin-β specific linker by Jodal et al., exemplified with exendin-4, also failed in vivo. The enzyme Meprin-β is highly expressed on the kidney brush border membrane, and there are well-known amino acid sequences that are cleaved by it [110]. The group showed highly promising results with an in vitro test but could not observe a renal reduction effect in tumor-bearing mice [111].

The tripeptide MVK (methionine–valine–lysine) was recently used by Arano et al. as a highly effective cleavable linkage for a Fab’ fragment of a mAb against c-kit, radiolabeled with Ga-67 labeled S-2-(4-isothiocyanatobenzyl)-1,4,7-triazacyclononane-1,4,7-triacetic acid ([^67/68^Ga]Ga-NOTA-p-SCN). C-kit is a tyrosine kinase receptor, which is an attractive target on cancer cells. They considered methionine–valine as a cleavable amino acid sequence for the brush border membrane enzyme neutral endopeptidase (NEP) [112]. The produced substrate [^67/68^Ga]Ga-NOTA-M showed fast excretion from the coated vesicle of the renal cells into the urinary tract [113]. The fragment with the cleavable linker showed a significantly lower kidney uptake up to 24 h in comparison to the directly coupled Fab’ without affecting tumor uptake. The highest kidney uptake reduction could be achieved after 6 h with ~85% reduction. The group also showed that all radiolabeled compounds exhibited high stability in serum [114]. These results led to several further promising studies with this approach.

Arano et al. examined in a detailed mechanistic study that the cleavage may not only happen during internalization in coated pits but also in coated vesicles. The radioactive metabolites are expected to be excreted to urine if they are cleaved during internalization, whereas for radiometabolites produced in coated vesicles, a distribution partially to the blood as well as excretion to the urine is proposed. The redistribution of the radiometabolites may result in higher liver uptake [115].

The approach was adapted from the same group to ^99m^Tc- and ^188^Re-labeled Fab’ fragments. They used a new cleavable linker GFK (glycine-phenylalanine-lysine) and renal secreted Tc-99m and Re-188 chelators were liberated. It was shown by blocking experiments with phosphamidon that NEP was the acting enzyme. The expected metabolites were detected in the in vitro assay as well as in the urine of mice injected with the radiolabeled fragments. The kidney uptake could be reduced from 11.1 %IA/g from a non-cleavable control compound to 4.2 %IA/g of the cleavable derivative with retained tumor uptake [116].

The group of Zalutsky adapted the enzymatically cleavable linker approach to different targeting moieties. They exemplified it by using a radioiodinated derivative of a PSMA ligand with a GY (glycine–tyrosine) linker. In healthy mice, they observed a 1.4-, 2.8-, and 161-fold kidney uptake reduction with the enzymatic cleavable linker at 1 h, 4 h, and 24 h, respectively. In tumor-bearing nude mice, the kidney uptake was reduced 7-fold at 21 h, but was not significantly changed at the other time points. However, the tumor uptake was also 3-fold lower at the 21 h time-point [117].

They also used the strategy for sdAbs labeled with F-18 for assessing the status of oncological targets such as the human epidermal growth factor receptor 2 by PET. Therefore, they developed an F-18 labeling method that utilizes the trans-cyclooctene (TCO)-tetrazine (Tz)-based inverse-electron demand Diels–Alder reaction in combination with a renal brush border enzyme-cleavable GK (glycyl-lysine) linker in the prosthetic moiety. As a radiolabeling method, the [^18^F]AlF-NOTA-PEG4 group for residualization was used for the sdAbs. The tumor uptake in SKOV3-bearing mice was ~80% of the [^18^F]AlF-NOTA-PEG4 compared to an iodinated reference compound, but the tumor-to-kidney ratio could be improved 4-fold, because of a 5–6-fold reduced kidney uptake [118].

The same group examined [^18^F]-6-fluoronicotinoyl radiolabeled fragments in more detail. In addition to biodistribution, in vitro metabolism and affinity changes were also evaluated. [^18^F]-6-fluoronicotinoyl coupled via a PEG4 and Gly-Lys cleavable linker showed only limited improvement related to the tumor-to-kidney ratio, which was increased 2.5-fold after 3 h compared to the iodinated derivative without a cleavable linker, [119] whereas a site-specific coupling with a maleimide to a cysteine engineered at the C-terminus showed promising results. The tumor uptake was retained after 1 and 3 h compared to the iodinated reference, whereas the kidney uptake was reduced by a factor of 4- and 6-fold after 1 h and 3 h. This resulted in an increased tumor-to-kidney ratio of 3.3- and 4.5-fold for F-18 (1.2 ± 0.4 and 12.2 ± 3.5) compared to those for I-125 (0.37 ± 0.05 and 2.7 ± 0.7) at 1 and 3 h, respectively.

Also, peptides often suffer from high kidney uptake, which results in dosage limitations. Therefore, cleavable linkers were also employed for exendin-4. Exendin-4 is a peptide that binds with high affinity to the GLP-1R and is routinely used to detect insulinomas [120] and focal congenital hyperinsulinism [121]. The kidney uptake reduction of [^68^Ga]Ga-NOTA-MVK-Cys40-Leu14-exendin 4 could be significantly reduced from 94.2 ± 9.8 %IA/g for the reference compound without the cleavable linker to 30.8 ± 3.8 %IA/g after 2 h, without affecting the tumor uptake [108]. Although this significantly reduced kidney uptake may improve the detection of lesions near the kidneys, the kidney dose is still too high for therapeutic application. The group tested different potential cleavable linkers to further reduce the kidney uptake. Different MXK (X = V, G, F and W) linkers were evaluated. No further optimization could be achieved [122].

An optimized derivative with a dual MVK approach was evaluated for [^111^In]In-FnBPA5.1, a peptide derived from bacterial adhesins that binds to relaxed fibronectin present in the tumor microenvironment. Valpreda et al. compared the direct coupling to a single MVK and two double MVK (MV-amBn-MVK and MVK(Me)2-amBn-MVK) motifs. The single MVK compound showed a decrease of 43%, resulting in a 53 %IA/g uptake in the kidneys at 24 h in comparison to the reference compound without a cleavable linker. A significantly better reduction with the dual cleavable linker of 63% and 54% could be achieved. But, a significant reduction of tumor uptake for the dual MVK linkers was also observed. However, the tumor-to-kidney ratios at 24 h p.i. were improved for all MVK-based FnBPA5.1 conjugates, with MV-amBn-MVK displaying an almost 2-fold increase compared to the compound without a cleavable linker [123]. Although this is a promising result, it is still not good enough for therapeutic application.

Arano et al. have written a detailed review about renal cleavable linkers [124].

In summary, cleavable linkers are a highly interesting approach to significantly reduce kidney uptake. However, further optimization is still necessary, and the mechanism of in vivo cleavage and how metabolites are processed has not been fully understood. Additionally, there is no proof of concept (yet) that this approach provides benefits for therapeutic application, where kidney uptake is often a main problem for small proteins and peptides.

## 7. Albumin-Binding Radioligands

The protein albumin circulates in the bloodstream and, owing to a mechanism that protects the protein from being degraded intracellularly, has a half-life of up to three weeks [125]. Initially, albumin was used as a carrier for drug delivery to prolong the circulation time of fast-clearing drugs. These long-acting drugs were designed to treat, for instance, diabetes, cancer, or infectious diseases [126]. In the context of radioligands, ligands comprising an albumin-binding moiety demonstrate extended circulation time, which could enhance tumor uptake. Extensive research has been conducted to improve the pharmacokinetic properties of radiolabeled compounds. Many studies aimed to enhance tumor uptake to increase the tumor absorbed dose and therapy efficacy, whereas others primarily focused on lowering kidney uptake to enable radioligand therapy in the first place and prevent nephrotoxicity. Examples will be provided to illustrate the leverage that an albumin-binding entity can have on the pharmacokinetics of a radioligand and whether it makes a radioligand suitable for therapy or not.

Fast clearance of folate analogues, targeting the folate receptor, leads to low uptake of these analogues in the tumor, accompanied by high uptake in the kidneys. Hence, the development of folate-receptor-targeted therapy is obstructed by this high renal uptake, which is partly due to folate receptor expression in the proximal tubule cells [127]. Based on the findings of Neri’s group [128], Müller et al. introduced the albumin binder 4-(p-iodophenyl)butyric acid (Figure 6) to prolong the circulation time of their ^177^Lu-labeled folate analogue in order to achieve a better tumor-to-kidney ratio. Indeed, they found increased tumor uptake for the folate with the albumin-binding entity (~18 %IA/g) compared to the folate lacking the albumin binder (~8 %IA/g) at 4 h after injection. Furthermore, retention in the tumor improved considerably (~11 %IA/g for folate with albumin binder vs. ~3 %IA/g for folate without albumin binder at 72 h after injection). Conversely, kidney uptake was reduced by more than 60% at 4 h after injection (~28 %IA/g for the albumin-binding compound and ~74 %IA/g for the compound without the albumin binder). Accordingly, a better tumor-to-kidney ratio was obtained, which enabled the use of radiolabeled folates in a preclinical therapy study for the first time [127]. Injection of 20 MBq of this ligand resulted in an absorbed tumor dose of ~36 Gy. However, it appeared that the absorbed kidney dose (~69 Gy for mice) was still very high, and together with prolonged retention in blood, further examination of the radiation safety of this new compound was warranted. In a follow-up therapy study using the same albumin binder-carrying compound, it was shown that nephropathy occurred for different doses tested, although to a lesser extent than the compound without the albumin binder [129].

Another application in which alleviation of kidney uptake is essential to allow TRT but also improve current diagnostics is GLP-1R targeting using radiolabeled exendin. Four new ^111^In-labeled exendin-analogues, containing the albumin binder 4-(p-iodophenyl)butyric acid at different positions in their sequence, were assessed. The compound with the best tumor-to-kidney ratio exhibited a tumor uptake of ~24 %IA/g and a kidney uptake of ~63 %IA/g 4 h after injection, whereas the reference compound without the albumin binder had a tumor uptake of ~9 %IA/g and kidney uptake of ~161 %IA/g [130]. Whether the promising improvements are sufficient to prevent radiation damage and enable TRT using radiolabeled exendin needs to be elucidated in future studies. Likewise, the kidneys are the dose-limiting organs for TRT of somatostatin-receptor expressing lesions. Even though renal uptake of radiolabeled somatostatin analogues can be inhibited significantly by co-administration of amino acids, loss of kidney function can emerge. Additionally, a prolonged circulation time could be beneficial to improve the overall tumor-to-organ ratios. In this respect, Rousseau et al. designed a DOTA-TATE derivative containing 4-(p-iodophenyl)butyric acid. In contrast to the above, for this ligand, increased kidney retention outweighed the increased tumor retention, rendering this ligand not suitable for therapy [131].

As research in the field of PSMA has emerged exponentially in nuclear medicine in the last few years, it is not surprising that so too is the introduction of albumin-binding entities to PSMA. The rationale for using albumin-binding PSMA ligands lies in the 30% of patients that do not respond to current PSMA-targeted radioligand therapies. As one hypothesis is that this could be due to an inadequate absorbed dose to the tumor, research is focused on increasing the dose delivered to the tumor, hence, increasing tumor uptake. By and large, most published studies that implemented an albumin-binding moiety were successful in enhancing the tumor uptake of a variety of PSMA ligands. But, conflicting with the results obtained for exendin and folate analogues, the renal uptake was also increased after implementing an albumin-binding moiety. We would like to portray some of the research, as these give more insight into the benefits of albumin-binding moieties, but also the undesired effects.

A head-to-head comparison of a ^177^Lu-labeled-PSMA-targeting ligand containing 4-(p-iodophenyl)butyric acid (CTT1403) in PC3-PIP tumor-bearing mice revealed higher tumor uptake (~47 %IA/g at 72 h p.i.) and longer retention in the tumor than the compound minus the albumin-binding entity (~1 %IA/g at 72 h p.i.), but also, kidney retention was increased (~48 %IA/g and ~7 %IA/g at 72 h p.i., respectively) [132]. A follow-up therapy study in mice has been conducted [133], and a phase I clinical trial is ongoing (NCT03822871). Another preclinical study in mice bearing PC3-PIP tumors compared the albumin binding, ^177^Lu-labeled PSMA-ALB-02 (containing 4-(p-iodophenyl)butyric acid) with PSMA-617. An increased tumor uptake led to an almost 2-fold increase of the AUC for PSMA-ALB-02 in comparison to PSMA-617. However, a 10-fold increase of the AUC was also found for the kidneys due to an increased uptake of PSMA-ALB-02 [134]. An idem trend was observed for ^177^Lu-labeled HTK01169, a PSMA-617 derivative containing N-[4-(p-iodophenyl)butanoyl]-Glu, which showed an increased tumor and kidney retention compared to PSMA-617. Tumor and kidney uptake were ~56 %IA/g and ~125 %IA/g for HTK01169, against ~11 %IA/g and ~0.6 %IA/g for PSMA-617 at 24 h p.i. [135]. The group of Babich developed a PSMA-targeting ligand, coupled to a p-(iodophenyl)acetic acid moiety, and after labeling with I-131, they showed a tumor uptake of ~6 %IA/g and a low kidney uptake of ~2 %IA/g at 24 h p.i. in LNCaP-bearing mice [136]. To allow labeling of this compound with a radiometal, they attached the chelator p-SCN-Bn-DOTA. Additionally, PEG linkers with different lengths were introduced between the PSMA-binding domain and the albumin-binding domain to make both domains function more effectively. For all new compounds, both tumor and kidney uptake were higher than for the reference compound PSMA-617 [137]. In a follow-up study, the compound was further modified by an additional PEG linker. This resulted in a better tumor-to-kidney ratio in LNCaP-tumor bearing mice [138]. According to them, their compound showed reduced kidney uptake (~7 %IA/g at 24 h p.i.), increased tumor uptake (~35 %IA/g at 24 h p.i.), and longer tumor retention relative to other PSMA ligands, like PSMA-617, CTT1403, PSMA-Alb-02, and PSMA-Alb-56 [139]. A head-to-head-comparison of these compounds in the same tumor model is needed to confirm whether this statement is correct. Finally, they substituted the DOTA chelator with Macropa (RPS-074), aiming for more efficient labeling with Ac-225 and better in vivo stability of the radiolabeled compound. Therapy studies in mice bearing LNCaP tumors showed an anti-tumor effect, but no information regarding toxicity was provided. Extrapolated dosimetry studies estimated an absorbed kidney dose of 84 mSv/MBq in human patients, which is a 10-fold lower absorbed kidney dose compared to [^225^Ac]Ac-PSMA-617 (700 mSv/MBq), but data on bone marrow and tumor dose were not reported [139]. However, it must be noted that the absorbed kidney dose of [^225^Ac]Ac-RPS-074 was based on preclinical data, whereas the dose of [^225^Ac]Ac-PSMA-617 was based on [^177^Lu]Lu-PSMA-617 data in human patients [140]. Nevertheless, the study sequence that they have made underlines the efforts that are needed to find a ligand with optimal pharmacokinetic properties to enable therapy studies. Additionally, other groups have combined albumin-binding moieties with linkers to improve pharmacokinetics [134,141,142,143,144]. Recently, a PSMA ligand with two 4-(p-iodophenyl)butyrate residues and two PSMA-binding moieties was radiolabeled with Ac-225, and evaluation showed that elevated tumor uptake and longer renal retention were inferior to the compound with one PSMA and one albumin-binding residue [145].

In addition to the albumin binder p-(iodophenyl)butyric acid, a variety of other albumin-binding entities have been explored. Umbricht et al. synthesized two novel PSMA ligands: PSMA-ALB-53 (4-(p-iodophenyl)-based albumin binder) and PSMA-ALB-56 (p-(tolyl)-based albumin binder (Figure 6)). They found similar AUCs of tumor uptake for both ^177^Lu-labeled ligands. On the contrary, the AUC for kidney uptake was 3-fold lower for PSMA-ALB-56 than for PSMA-ALB-53, leading to a much more favorable distribution profile. The authors declared that the decreased renal uptake could be explained by PSMA-ALB-56 clearing faster from the blood, as a result of less binding affinity to albumin compared to PSMA-ALB-53 [146]. The promising results obtained for PSMA-ALB-56 led to a clinical study in mCRPC patients [147]. Unfortunately, the suboptimal tumor-to-blood ratio, accompanied by elevated doses to the bone marrow and kidneys, made them go back to the bench to develop new analogues with a more favorable distribution profile. Accordingly, the same group investigated several PSMA ligands comprising the albumin binder isobutylphenyl propionic acid (ibuprofen) (Figure 6) and used a variety of differently charged amino acids to link the albumin-binding moiety to the ligand. It is interesting that, for the ligand with the lowest binding affinity to plasma proteins, the lowest kidney retention in vivo was observed ([^177^Lu]Lu-Ibu-DAB-PSMA). Apparently, the combination of the positively charged amino acid (diaminobutyric acid (DAB)) with the albumin-binding moiety had a beneficial effect on the distribution profile. To illustrate, [^177^Lu]Lu-Ibu-DAB-PSMA exhibited a kidney uptake of ~6 %IA/g at 24 h p.i., whereas the kidney uptake of [^177^Lu]Lu-Ibu-PSMA (which had a higher albumin-binding affinity) was ~16 %IA/g [134,148]. In a subsequent therapy study on mice, it was reported that the increased tumor retention of ^177^Lu-labeled Ibu-DAB-PSMA and PSMA-ALB-56 resulted in better therapeutic efficacy compared to PSMA-617. Additionally, Ibu-DAB-PSMA did not show any hematological side effects, in contrast to PSMA-ALB-56. Hence, in this mouse study, the ligand with the weaker albumin binder (Ibu-DAB) was superior to the ligand with the stronger albumin binder (p-tolyl) and even superior to PSMA-617 [149]. In the next study, two stereoisomers of ibuprofen were compared, as S-ibuprofen is pharmacologically more active than R-ibuprofen [150]. It is striking that in this study the stronger albumin binder (SibuDAB), which had slower blood clearance, had an almost 2-fold lower kidney retention than the weaker albumin binder (RibuDAB) (AUC of 446 ± 48 %IA/g*h vs. 847 ± 80 %IA/g*h). This is in contrast to previous studies [142,146], where the stronger albumin-binding compound had, in fact, higher kidney retention (2430 ± 121 %IA/g*h for PSMA-ALB-53 and 809 ± 43 %IA/g*h for PSMA-ALB-56). Different parameters, like metabolic stability or charge, may be a more plausible reason for the better distribution profile of SibuDAB, rather than the albumin-binding affinity alone. Because of higher in vivo stability and lower kidney uptake of [^177^Lu]Lu-SibuDAB-PSMA, this ligand was translated to an alpha-emitting therapeutic. [^225^Ac]Ac-SibuDAB was superior to [^225^Ac]Ac-PSMA-617 in terms of therapeutic efficacy, and in the time window tested, no increased toxicity was found, although future (pre)clinical studies are needed to determine absorbed doses and possible long-term toxicity [151]. Another study conducted a head-to-head comparison of ^86^Y-labeled PSMA-617, either coupled to the moderate albumin-binding 4-(p-iodophenyl)butyric acid (IP-PSMA-617) or Evans Blue (Figure 6) (EB-PSMA-617). They observed higher tumor retention for EB-PSMA-617 than for IP-PSMA-617 or PSMA-617 [152]. Although kidney retention was also much higher, a first clinical study was assessed in which mCRPC patients received a single low dose of 177Lu-labeled EB-PSMA-617 (0.80–1.11 GBq) or PSMA-617 (1.30–1.42 GBq). The albumin-binding ligand was shown to have prolonged tumor retention and better therapeutic efficacy than PSMA-617. Because of slower blood clearance, higher absorbed doses to other organs, like the bone marrow and kidneys with a 6-fold increase, were also observed [153]. A more elaborate study is needed to confirm the findings and safety of this therapeutic compound.

Bandara and colleagues conjugated Evans Blue to DOTA-TATE (DOTA-EB-TATE) and demonstrated higher tumor retention, coinciding with better therapeutic efficacy, compared with DOTA-TATE (both labeled with Lu-177). However, accumulation in all organs increased drastically, but as data on renal uptake or toxicity studies are missing, the safety of this ligand is uncertain [154]. Similar results were exhibited when DOTA-EB-TATE was labeled with Y-90, although they observed no hematological deviations between the treatment group and the untreated group of mice [155]. These findings encouraged the researchers to employ clinical studies in patients with advanced neuroendocrine tumors. The first study evaluated the dosimetry and safety of administering 0.35–0.70 GBq of [^177^Lu]Lu-DOTA-EB-TATE in comparison to 0.28–0.41 GBq of [^177^Lu]Lu-DOTA-TATE. The estimated effective dose was elevated for almost all organs. A 7.9-fold increased radiation exposure to the tumor was found, but also, a 3.2-fold and 18.2-fold exposure to the kidneys and bone marrow, respectively [156]. In the second study, it was investigated if a low dose of [^177^Lu]Lu-DOTA-EB-TATE (0.65 ± 0.06 GBq) could already induce a response. In 3/4 patients, a partial response was shown, equal to treatment with 3.98 ± 0.17 GBq of [^177^Lu]Lu-DOTA-TATE, where 2/3 patients had a partial response [157]. For both studies, no short-term side effects were observed, indicative of a safe treatment regimen. Comprehensive studies should elaborate on the long-term safety of this compound.

Recently, fibroblast activation protein (FAP) targeting using FAP inhibitors (FAPIs) in combination with either Evans Blue [158] or 4-(p-iodophenyl)butyric acid [159] was also assessed in preclinical studies. Overall, prolonged blood and tumor retention were reported, but also higher renal retention. Interestingly, the ligand with 4-(p-iodophenyl)butyric acid displayed higher blood retention, whereas the ligand with Evans Blue displayed higher kidney retention. Nevertheless, tumor growth inhibition by both compounds was shown, with (almost) negligible side effects. Future dosimetry and toxicity studies have to reveal the clinical feasibility of these ligands.

A detailed review about the use of albumin binders to improve cancer radioligand therapies has been written by Lau et al. [160].

With only few examples, we have illustrated that the incorporation of an albumin-binding moiety is indeed able to diminish kidney uptake and enhance tumor uptake. However, in most studies, kidney uptake was actually elevated, and additional modifications of the ligand were needed to reverse the high kidney uptake. So, it seems that the addition of an albumin-binding moiety is merely not enough to obtain the most optimal distribution profile. In addition, uptake in other dose-limiting organs, for instance, the bone marrow (higher radiation exposure due to prolonged blood circulation) and other target-expressing normal tissues, have to be carefully examined. It is not clear why, for some ligands, kidney uptake benefits from the addition of an albumin-binding entity, whereas for others, it is not advantageous. A stronger albumin-binding affinity impedes the release of the compound from albumin, leading to less enhanced uptake in the target organ. Conversely, too low an affinity will result in a shorter circulation time but does not necessarily lead to lower kidney uptake (as was shown for PSMA-ALB-56 and Sibu-DAB). A delicate balance between the affinity of the albumin-binding entity and affinity for the target receptor has to be found in order to obtain an optimal distribution profile.

## 8. Discussion, Conclusions, and Future Directions

Several attempts have been made to prevent renal damage during TRT by developing methods that reduce renal uptake of peptides and small proteins. Many of the published methods show a significant effect but suffer from drawbacks, as summarized in Table 2. For instance, a molecular change that reduces uptake of a radioligand in the kidneys may decrease affinity for the target in the tumor, which restricts its utility. Additionally, prevention of kidney uptake by redirecting the distribution of the radioligand to other organs like the liver or/and blood (longer blood circulation time) may lead to additional or altered toxic side effects, which puts other organs at risk as dose-limiting organs. Furthermore, for all strategies discussed, the effect on kidney uptake is not uniform for every radioligand. One reason for this is that kidney physiology is still not well-understood, and the effects of approaches are sometimes unpredictable. Different radioligands can exhibit distinct mechanisms of reabsorption, indicating that different approaches are needed to reduce kidney uptake. Potentially, a combination of different methods may be necessary to achieve sufficient reduction, although this part of the field is still not well-evaluated. Very often, it is a trial-and-error approach.

Another factor that challenges the successful development of radioligands is evaluating the effects of a method to reduce radioactivity accumulation in the kidneys in an accurate way. Probably the main factor to be considered when evaluating the different methods to diminish kidney uptake is determining the renal retention over time, rather than at a single timepoint. Specifically, in view of therapeutic radioligands, dosimetry is indispensable. The activity exposure over time determines the maximum dose of a radiotherapeutic. Measuring radioactivity at a certain timepoint does not necessarily provide information about residualizing activity in the renal tubules; a ligand rapidly clearing from blood shows high radioactive accumulation in the kidneys at early timepoints, which may appear dramatic if measured at this timepoint, while the high dose delivered to the kidneys might only be for a short duration. Therefore, thorough evaluation of radioligands is necessary to provide information about residualizing activity in the kidneys.

In addition, tumor uptake greatly influences the radioligand dose that will be administered to a patient or test animal. High tumor uptake or increased tumor retention may allow a lower administered radioactive dose that is still sufficient to treat the tumor, with limited radiation dose to the kidneys. Therefore, a good trade-off between the tumor and kidney uptake should be examined thoroughly, as for other dose-limiting organs such as the bone marrow.

The factors mentioned make the development of new radioligands with limited kidney uptake a challenging mission. For most radioligands, the reduced renal uptake obtained is not sufficient for therapeutic application. Still, few renal reduction methods have made it to clinical applications (e.g., [27,39,161]). Among these, the most used strategy is blocking the kidney uptake of the radioligand by the co-infusion of an arginine/lysine solution, as applied during targeted radiotherapy with somatostatin receptor ligands, including Lutathera. The use of I-131, with non-residualizing activity in the kidneys for the labelling of small proteins, seems to be another interesting approach. After a successful clinical dosimetry and imaging study, the radioiodinated sdAb targeting Her2/neu is currently being examined in a dose-escalating study. A third substance in a dose-escalating study is being examined for the treatment of medullary thyroid carcinoma. This is a peptide ligand in which a sequence of six L-glutamic acids, which were responsible for high kidney uptake, was replaced by a sequence of the D-isomers.

In conclusion, different approaches for the reduction of renal kidney uptake are available, but some are still poorly understood, and their effects are insufficient. There will not be a ‘one size fits all’ strategy to reduce kidney uptake. Therefore, we see great potential in persisting with current efforts and in exploring the combination of different kidney uptake reduction approaches.

## Figures and Tables

**Figure 1 pharmaceuticals-17-00256-f001:**
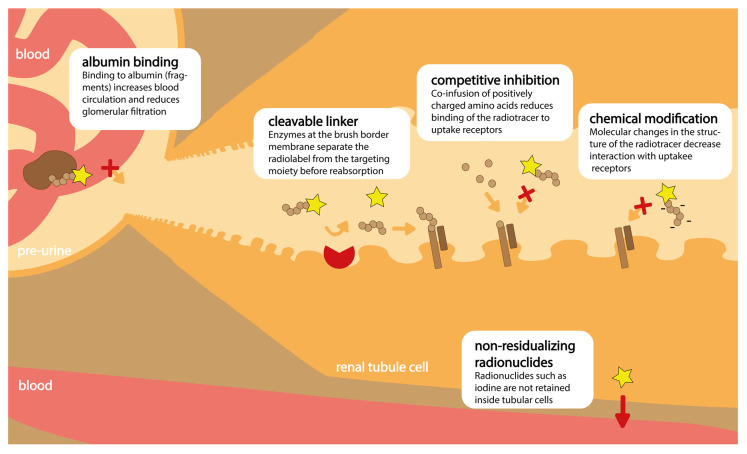
Overview of mechanisms discussed in this review for the reduction of kidney uptake of radioligands. Brown dots represent amino acids assembled into a radioligand, yellow star represents radiolabel.

**Figure 2 pharmaceuticals-17-00256-f002:**
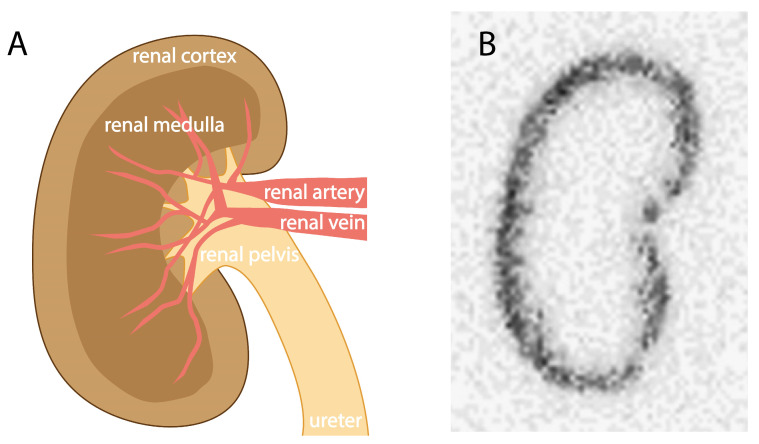
(**A**) Schematic representation of a cross-section of a kidney, showing the renal cortex (light brown) and medulla (dark brown). (**B**) Autoradiogram of a rat kidney showing accumulation in the renal cortex after administration of radiolabeled minigastrin. This research was originally published in JNM. Gotthardt et al. Indication for Different Mechanisms of Kidney Uptake of Radiolabeled Peptides. *J. Nucl. Med.*
**2007**, *48*, 596–601. © SNMMI [7].

**Figure 3 pharmaceuticals-17-00256-f003:**
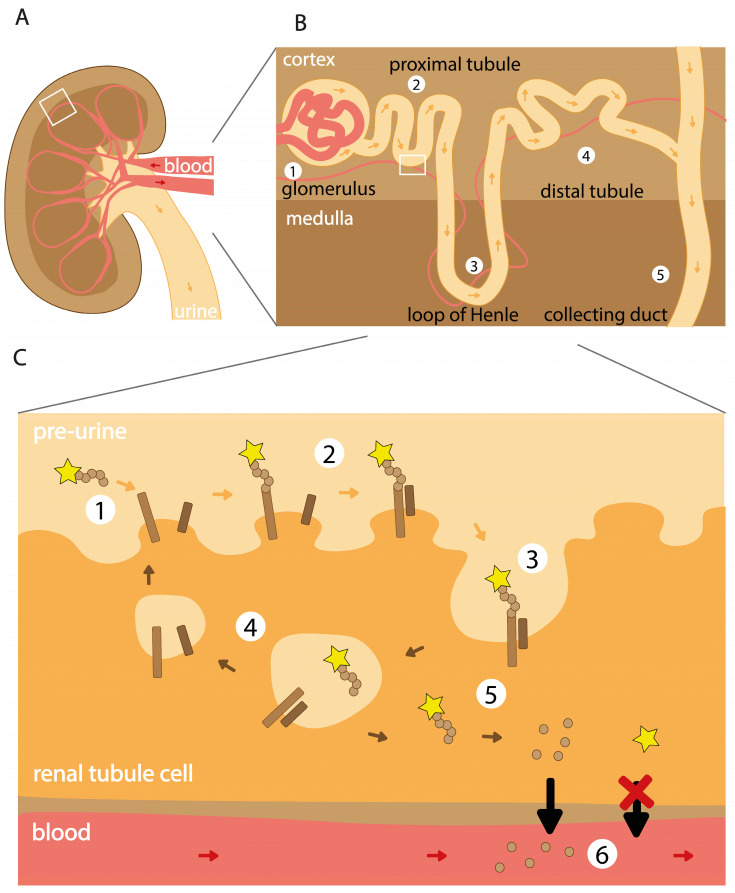
Schematic representation of kidney physiology at different magnification levels and the mechanism of accumulation of radioactivity in the kidneys. (**A**) Schematic representation of a cross-section of a kidney. The white square indicates the zoom frame of figure (**B**). (**B**) Schematic representation of a nephron. In renal excretion, dissolved ions and biomolecules are filtered in the glomerulus (1), after which the formed pre-urine travels through the proximal tubule (2) where reabsorption of organic compounds and water takes place. In the loop of Henle (3), water and subsequently ions are reabsorbed, after which additional reabsorption of water and ions can take place in the distal tubule (4). Urine reaches the collecting duct (5) and is transported towards the bladder. The white rectangle indicates the zoom frame of figure (**C**). (**C**) Schematic representation of a proximal tubule cell showing the binding of a radioligand (brown dots with yellow star) to megalin (light brown bar) on the apical site of the membrane (1), followed by association with cubilin (dark brown bar) (2). Subsequent endocytosis (3) internalizes the megalin–cubilin–radioligand complex into the tubular cell. The receptors in the endosome are recycled back to the apical membrane (4), while the radioligand is degraded into its biomolecular building blocks after fused with the lysosome (5). Where the radionuclide–chelator complex (yellow star) is trapped inside the tubular cell, on the basolateral side of the tubule cell, biomolecular building blocks (brown circles) are transported towards the blood circulation for re-use (6).

**Figure 4 pharmaceuticals-17-00256-f004:**
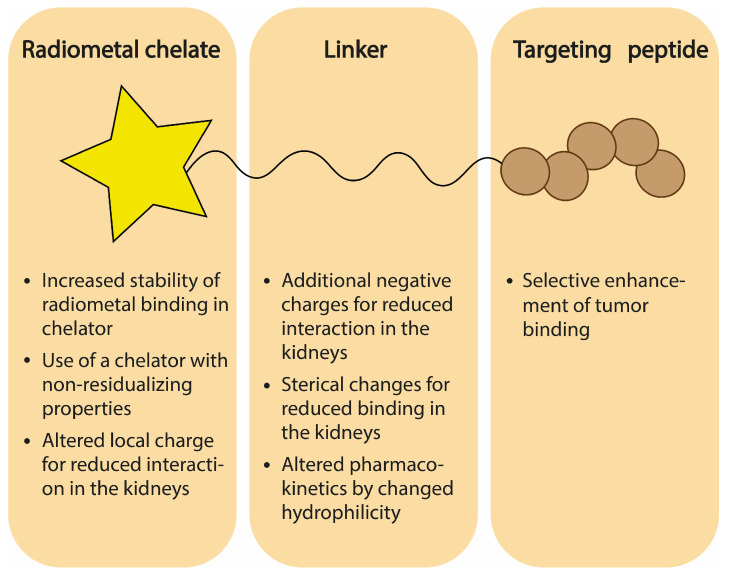
Overview of structural changes regarding the chemical design of radioligands for reduced uptake in the kidneys.

**Figure 5 pharmaceuticals-17-00256-f005:**
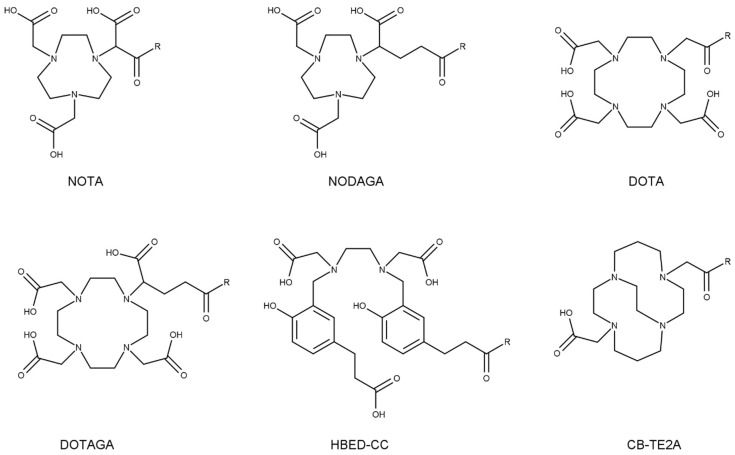
Chemical structures of several chelators discussed in this chapter. R represents conjugation site to (the linker of a) targeting ligand.

**Figure 6 pharmaceuticals-17-00256-f006:**
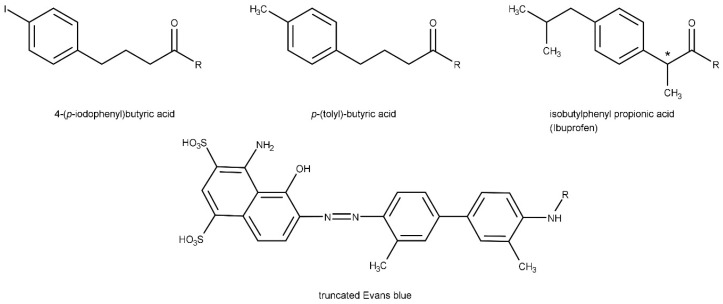
Chemical structures of several albumin-binding entities that have been discussed in this chapter. * indicates stereocenter.

**Table 1 pharmaceuticals-17-00256-t001:** Overview of renal cleavable linkers, as discussed in this section.

Sequence	One Letter Code	Enzyme
Gly-Lys	GK	Carboxypeptidase M
Met-Val-Lys	MVK	Neutral endopeptidase (NEP)
Gly-Phe-Lys	GFK	Neutral endopeptidase (NEP)
Gly-Tyr	GY	Carboxypeptidase M

**Table 2 pharmaceuticals-17-00256-t002:** Overview of strategies used to reduce uptake and retention of therapeutic radioligands.

Strategy	Advantages	Limitations and Disadvantages	Development Phase
Use of non-residualizing radionuclides	Robust effect on kidney uptake.	Not compatible with radiometals, which include most commonly used therapeutic radionuclides. Challenging radiosynthesis.	Clinical trials
Co-administration with competitive inhibitors	Compatible with all therapeutic radionuclides. No need for alteration of radioligand design.	Effect on kidney uptake not always sufficient for treatment application. Potential allergic reactions.	Applied in the clinic
Adaptation of the chelator or linker design on molecular level	Compatible with all therapeutic radionuclides. Modification in the early development phase.	Target-binding affinity often reduced. Effects are hard to predict.	Applied in the clinic
Cleavable linkers	Compatible with all therapeutic radionuclides.	Effect on kidney uptake not always sufficient for treatment application. Off-target cleavage can lead to reduced tumor uptake.	Preclinical development
Albumin binding	Compatible with all therapeutic radionuclides. Potential for enhanced tumor uptake.	Effect is hard to predict. Effect on kidney uptake not always sufficient for treatment application. Enhanced circulation time increases dose delivered to the bone marrow.	Clinical trials

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
