# Peer review of "Towards the Magic Radioactive Bullet: Improving Targeted Radionuclide Therapy by Reducing the Renal Retention of Radioligands"

_pharmaceuticals, 2024, doi:10.3390/ph17020256_

Round 1

Reviewer 1 Report

Comments and Suggestions for Authors

This review summarizes the approach to reduce the renal accumulation of radioligands, focusing on therapeutic applications. However, the pioneering studies of each method usually used diagnostic radioisotopes, and these studies should be presented first.

The following are some of the concerns that should be addressed.

3. radionuclide dependent kidney uptake reduction.

This section compares [131I]SGMIB-sdAbs with 177Lu-labeled sdAbs. Whether sdAbs are internalized after binding to tumor cells and the bifunctional chelating reagent used for 177Lu labeling should be clearly described. Furthermore, SGMIB is a labeling agent residualized within the cells much longer than SIB. However, it is difficult to understand why the [131I]SGMIB-sdAbs showed higher tumor accumulation and faster elimination rates of radioactivity from the kidney than the 177Lu-labeled ones. The reason for this should be clearly stated. To cite these two labeled agents as examples of radionuclide-dependent kidney uptake reduction seems inappropriate.

4. Competitive inhibition in the proximal tubules.

The effect of charged amino acids and gelofusine on reducing the renal accumulation of labeled peptides has been described. The relationship between these compounds and megalin/cublin should be summarized. The reasons for radioligands that provided exceptional results should also be described with potential reasons.

5. Chemical design of the radioligand

To the best of the reviewer's knowledge, the first study on the charge-modification of radiolabeled labeled peptides was that of Akizawa et al. on 111In-DTPA-labeled TOCs. These studies should be described, and subsequent investigations, including those by the linker, should be followed as extensions of the work.

The stability of radiometal chelate is a fundamental issue, and this should be clearly stated at the beginning of the text, not in the middle of a chapter.

6. Cleavable linkage

Indeed, Dean et al. were the first to apply an ester bond to reduce the renal radioactivity of 99mTc-labeled Fab'. However, it should also be noted that this method was first proposed by Prof. Meares' group at UC Davis to reduce the hepatic accumulation of 111In-labeled intact IgG (facilitating excretion of radioactive metabolites from liver lysosomes). This concept was applied to the kidney by Dean et al. to facilitate the elimination of radioactivity (99mTc-MAG3) from the renal lysosomes to the urine.

The reduction of renal radioactivity using the brush border membrane enzyme is an alternative approach for reducing the accumulation of radioactivity in the kidneys. This strategy involves the release of radiocatabolites at the renal brush border (not from the renal lysosomes) from antibody fragments. Thus, the description of this approach should be clarified from the approach mentioned above (lysosomes) not to confuse the readers.

7. albumin-binding radioligands

This method was initially designed to improve accumulation in tumors, and as a secondary effect, renal accumulation is reduced when the binding affinity to albumin is appropriate. The relationship between the binding affinity to albumin and the tumor and the renal accumulation should be described.

Comments on the Quality of English Language

No issues found

Reviewer 2 Report

Comments and Suggestions for Authors

This is a very well written review article describing challenges related to renal clearances and possible causes and solutions to circumvent them. The paper focuses on small molecules and peptides primarily. To further enhance the paper, the authors can possible add a couple of sections related to antibodies and now different factors such as glycans, PI and etc affect clearance properties and possible ways to optimize them.

Reviewer 3 Report

Comments and Suggestions for Authors

he number of reviewed papers is sufficient to discuss the state of the art. However the text is long with few or no figures. This could be useful to readers who might be unfamiliar with this topic. Perhaps the rights for images or figures could be obtained from some key topics discussed.

Another improvement I would like to see is perhaps a table which summarizes the discussion section and highlights the main methods and some of the advantages or drawbacks. Citations/Refs could be used here too.

Comments on the Quality of English Language

Please see attached PDF for suggestions and comments. The English language used in this manuscript needs to be revised. It is obvious that this is authored by a non-native and makes reading difficult for native speakers. 

Round 2

Reviewer 1 Report

Comments and Suggestions for Authors

1. Page 6, lines 225-228. The revised statement is far from the common understanding: In general, an identical radiolabeled metabolite is produced after lysosomal proteolysis of parental antibody fragments/constructs in the tumor and kidney. As a result, directly radioiodinated and [*I]SIB-labeled antibody fragments/constructs internalized into tumor cells show low tumor and renal radioactivity levels, whereas [111In]In-DTPA-labeled ones exhibit high tumor and renal radioactivity levels. As the authors state, suppose SGMIB-labeled antibody constructs show different radioactivity levels in these tissues. In that case, the authors should describe the mechanism(s) underlying the different radioactivity levels and provide more references to support the statement.

2. Page 7, lines 303-308. The statements become more reliable when the authors add some references supporting the statements.

3. Section 6. Cleavable linkers. I read the paper by the Arano group. They mentioned more clearly that the generation of the radiometabolites would occur not only during the internalization process of the parental antibody fragments but also after the formation of coated vesicles (Bioconjugate Chem. 31 (11): 2618-2627, 2020); Nucl Med Biol 92: 159-155, 2021). Thus, the authors should revise the statements.
